# Screening of Candidate Housekeeping Genes in Uterus Caruncle by RNA-Sequence and qPCR Analyses in Different Stages of Goat (*Capra hircus*)

**DOI:** 10.3390/ani13121897

**Published:** 2023-06-06

**Authors:** Yumei Zhou, Xingchun Li, Xinyue Zhang, Minghui Li, Nanjian Luo, Yongju Zhao

**Affiliations:** 1College of Animal Science and Technology, Southwest University, Chongqing 400715, China; 2Chongqing Key Laboratory of Herbivore Science, Chongqing Key Laboratory of Forage and Herbivore, Chongqing Engineering Research Center for Herbivores Resource Protection and Utilization, Chongqing Herbivore Engineering Research Center, Chongqing 400715, China; 3School of Preclinical Medicine, Zunyi Medical University, Zunyi 563000, China

**Keywords:** uterus caruncle, the candidate housekeeping genes, RNA-sequence, goat

## Abstract

**Simple Summary:**

The uterus is an important organ in the reproductive system of most female mammals. The normal growth and development of ruminant uterus caruncles are crucial to maintain gestation and fetal health in goats. The aim of this study was to screen a set of candidate housekeeping genes (HKGs) in *Carpra hircus* uterus caruncles. Four programs and one comprehensive index were applied to assess the stability of 22 selected HKGs expression during non-pregnant and pregnancy stages. The most and least stable HKGs were used to normalize the target genes expression of *SPP1*, *VEGFA*, and *PAG8*. Traditional reference genes were not suitable for target gene normalization, while *PPIB* and *EIF3K, POP4, PPIA,* and *SDHA* showed the least variation and were recommended as the best HKGs during the nonpregnant stage and the whole stages of goat uterus caruncle tissue, respectively. This study found the suitable HKGs in uterus caruncle tissues at different stages of non-pregnancy and pregnancy.

**Abstract:**

The uterus is a critical pregnancy organ for mammals. The normal growth and development of ruminant uterus caruncles are crucial to maintain gestation and fetal health in goats. Quantitative real-time polymerase chain reaction (qRT-PCR) is a reliable tool to study gene expression profiling for exploring the intrinsic mechanism underlying the conversion process of uterus caruncle tissue. However, the candidate housekeeping genes (HKGs) are required for normalizing the expression of function genes. In our study, 22 HKGs were selected from analyzing transcriptome data at non-pregnancy and pregnancy processes and previous reports about HKGs in goat tissues. We assessed them for expression suitability in 24 samples from uterus tissues at 15 non-pregnant days (Stage 1), early (Stage 2), and medium-later pregnant days (Stage 3). The expression stability of these genes was evaluated by using geNorm, Normfinder, Bestkeeper, and Delta Ct algorithms and, comprehensively, by ReFinder. In addition, the most and least stable HKGs were used to normalize the target genes expression of *SPP1*, *VEGFA*, and *PAG8*. It was found that traditional reference genes, such as *ACTB* and *GAPDH*, were not suitable for target gene normalization. In contrast, *PPIB* selected from RNA sequencing data and *EIF3K* selected from previous references showed the least variation and were recommended as the best HKGs during the nonpregnant stage and the whole stages of goat uterus caruncle tissue, respectively. It is the first time the HKGs genes in uterus during the non-pregnant day and throughout the total pregnancy have been explored. These findings found suitable HKGs in uterus caruncle tissues at various stages of non-pregnancy and pregnancy; these can be useful for gene expression studies to reveal the molecular mechanisms of uterus development in goats.

## 1. Introduction

The uterus, a place where the embryo is implanted and continues to develop, is necessary for the reproduction of mammals. Uterus caruncles are a special structure located on the endometrium of domestic ruminants. During the process of pregnancy, uterus caruncles participate in placentation and form to become the maternal part of placenta [1]. Before and after pregnancy, the uterus caruncle develops from endometrial epithelial cells to connective tissue rich in blood vessels. This dynamic process indicates that the structure and cell types of the uterus caruncle undergo significant changes during the pregnancy.

Goat (*Capra hircus*) is becoming widely accepted around the world due to the increasing demand for the low cholesterol content and high nutritive value of goat meat [2]. Successful pregnancy and fetal health in goats are important safeguards for reproductive performance and are dependent on normal uterus function and placenta development. The hircine placenta is the cotyledon placenta, which divides into multiple button-shaped functional units, entrenched into maternal uterus caruncle with the incomplete uterine luminal epithelium, and achieves the synepitheliochorial placenta at gestation [3,4]. Although several physiological processes have been reported in ruminant uteruses [5,6], the exact molecular mechanisms remain largely unclear in uterus caruncle development in domestic animals. Gene expression analysis is a useful technique because it provides information on the regulation of uterus development.

Quantitative real-time polymerase chain reaction (qRT-PCR) is widely applied to quickly and accurately characterize the expression of target genes. To obtain the real expression changes of target genes, HKGs must be used to normalize cycle threshold (Ct) value from qRT-PCR. In theory, the expression of HKGs should not change in the investigated different tissues or in development stages. To obtain the candidate housekeeping genes, programs (including geNorm [7], Normfinder [8], BestKeeper [9], and Delta-Ct method [10]) have been invented to facilitate the statistical analysis of potential endogenous control genes base on the results of qRT-PCR. Furthermore, comprehensive tools have also been developed to compare tested HKGs, including RefFinder [11] and ComprFinder [12]. Meanwhile, high-throughput RNA sequencing provided a large transcriptional collection of all genes, and the fragments per kilobase of transcript sequence per millions of base pairs sequenced (PFKM) has been used to screen for new or stable reference genes with low coefficient variation [13,14].

To date, several studies have identified the candidate HKGs in uteruses of other species, but not goats. For instance, the combination of *YWHAZ*, *HPRT1*, and *HMBS* was suggested as the most stable reference group of genes in the ovarian and uterine tissues of laying hens under control and heat stress conditions [15]; *YWHAZ* was the best reference gene, which could be used as an accurate internal control gene in canine pyometra studies [16]. Moreover, there were also some studies carried out on tissues other than the uterus of goat. The *HMBS* and *B2M* were identified as the candidate housekeeping genes in hircine spleen, caecum, small intestine, lower lip, large intestine, trachea, and upper lip [14]. Several HKGs were also demonstrated in other tissues of goats, such as *SDHA* in skin [12], *EIF3M* in adipose tissue [13], and *PPIB* and *HMBS* in intramuscular fat [17]. However, the candidate housekeeping genes for use in the study of goat uteruses, especially uterus caruncle, are lacking.

In this study, we aimed to screen a set of candidate HKGs in *Carpra hircus* uterus caruncle. Twenty-two original candidate HKGs were selected by analyzing transcriptome data and referring to previous studies. The transcriptional expression levels of these genes were detected in uterus caruncle by qRT-PCR. Four programs and one comprehensive index were applied to assess the stability of HKGs expression during non-pregnant and pregnancy stages. These findings were useful in selecting HKGs to correctly analyze the expression of function genes in the uterus caruncle, which aids in understanding the molecular genetic mechanism of pregnancy in goats.

## 2. Materials and Methods

### 2.1. Ethics Statement

All goat experiments were performed in strict accordance with the recommendations of the Southwest University Institutional Animal Care and Use Committee (2019, No. GB14925-2010). All research involving animals was conducted according to the approved protocols of the Chongqing Key Laboratory of Herbivore Science, Chongqing, China.

### 2.2. Animals and Sample Harvesting

The research involving animals was conducted according to the approved protocols of the Chongqing Key Laboratory of Herbivore Science, Chongqing, China. The animals were from the breeding farm at Southwest University. They were subjected to a standard feeding management regime (DB50/T 510-2013) and housed in free semi-open folds. Sample procedures in this study were supervised and approved by the Institutional Animal Care and Use Committee of Southwest University. Through cesarean operation, uterus caruncle samples were collected at eight different non-pregnant and pregnant stages of Dazu Black goats (Figure 1a,b).

### 2.3. RNA Extraction and cDNA Synthesis

Total RNA is extracted from uterus caruncle tissues using Trizol^®^ reagent according to the manufacturer’s instructions (Invitrogen, Carlsbad, CA, USA). RNA concentration was measured by NanoDrop 2000 spectrophotometer (Thermo Fisher, Meridian, ID, USA), and the purity of RNA was determined by the ratio of OD260/280. All the ratio of OD260/280 of our RNA samples value 1.8–2.0. Each sample was reverse transcribed using All-In-One 5X RT MasterMix (Abm, Heidelberg, Germany, Cat#G592) with 2 μg total RNA.

### 2.4. Selection of Candidate Reference Genes

Transcriptome sequencing data of 9 Dazu Black goat uterus caruncle tissues were obtained using the paired-end sequencing technology on an Illumina NovaSeq 6000 platform. After assembly and annotation, the gene expression profiles and read counts of unique transcripts were converted into FPKM values on the platform BMKCloud (www.biocloud.net) (accessed on 15 March 2021), according to the formula FPKM = cDNA fragments/[mapped fragments (millions) × transcript length (kb)]. Based on the FPKM value, every gene in each transcriptome, the CV, and MFC were calculated using Microsoft Excel. The CV is defined as the ratio of the SDFPKM to the mean of the FPKM of all samples for one gene. The MFC, which is defined as the fold change between the largest and smallest FPKM values within 9 RNA-seq, was calculated. A potential HKG should have a relatively high FPKM value [12,18] and low CV and MFC values because these are the typical requirements for HKGs, along with a relatively high and low expression variation level. We also got several HKGs from prior studies [19,20,21] and HKGs of frequently used genes in addition to genes chosen from transcriptome sequencing data with ideal FPKM, CV, and MFC values. qRT-PCR was used to amplify all HKGs for later analysis and confirmation.

### 2.5. Quantitative Real-Time PCR (qRT-PCR) Analysis

Specific primers were designed using the Primer-Blast web tool (https://www.ncbi.nlm.nih.gov/tools/primer-blast/ accessed on 22 March 2021) based on the sequences of unigenes. The information on primers is shown in Appendix A. The qRT-PCR was carried out in a CFX96 Real-Time System (BIO-RAD, Hercules, CA, USA) with SsoAdvanced SYBR Green Supermix (BIO-RAD, #1725262). The qRT-PCR reaction system was performed in a 10 μL reaction volume with 5 μL of 2× qPCR MasterMix (Abm, Richmond, BC, Canada, Cat#G891), 4.5 μL cDNA template, and 0.25 μL each primer. The thermocycling conditions were performed starting with a 3 min enzyme activation step at 95 °C, followed by 40 cycles of 15 s at 95 °C for denaturation, and 1 min at 60 °C for annealing/extension. Melting curve analysis was performed at the end of each PCR program to monitor non-specific product formation.

### 2.6. Analysis of the Expression Stability for Candidate Reference Genes

The qRT-PCR Ct (Cycle Threshold) data were entered into four traditional algorithms, including geNorm [7], NormFinder [8], BestKeeper [9], and the Delta Ct method [10], to evaluate the stability of HKGs. Moreover, ReFinder [11] was used to obtain a comprehensive ranking of HKGs.

### 2.7. Validation of HKGs

To validate the HKGs stability results, the expression profiles of 3 well known goat uterus caruncle genes (*VEGFA*, *PAG8*, and *SPP1*) were investigated by performing qRT-PCR analysis of cDNA samples, in the same way as stated above. The expression levels of target genes were normalized using the 2^−∆∆CT^ method [22]. The primer details of these target caruncle genes have been provided in the Appendix A.

## 3. Results

### 3.1. Selection of HKGs Using RNA-Seq Data

Based on the FPKM value, the coefficient of variation (CV, %), and the maximum fold change (MFC), we obtained a total of 383 HKGs (Figure 2). KEGG (Kyoto Encyclopedia of Genes and Genomes) enrichment analysis indicated that most HKGs were involved in the ribosome and lysosome pathways (Appendix A). Five HKGs, including *MORF4L1*, *HNRNPL*, *PTPRA*, *ARF1*, and *PPIB*, were selected for further validation. We acquired several HKGs from prior studies [20] and HKGs of widely used genes (*GAPDH*, *SDHA*, *ACTB*, *YWHAZ*, and *18S*) in addition to genes chosen from transcriptome sequencing data. Table 1 contains all the details of the HKGs we obtained from transcriptome data for our research.

### 3.2. Amplification Specificity and Efficiency of the HKGs and Target Genes

A total of 25 primer pairs, including 22 HKGs and three target genes, were designed for qRT-PCR experiment. The specificity for each paired primer was validated by the melting curve analysis, which showed a single amplification peak (Appendix A). Each pair of primers had good specificity and amplification efficiency around 100%.

### 3.3. Expression Profiling of the HKGs

The expression of 22 HKGs was assayed in a sample of caruncle. The distribution of the mean Ct values (the average of three technical replicates in the same sample) is displayed in Figure 3a. The mean Ct values range from 19.80 to 32.87. We selected genes with a mean Ct value lower than 28 for our subsequent analysis. In other words, HKGs whose mean Ct value was greater than 28, including *POLR2A*, *TBP*, *PTPRA*, *18S*, and *UBC*, were excluded in further analysis. These findings suggest that gene expression varied from approximately 300 folds per change compared to the mean gene expression of all 17 genes (Figure 3b).

### 3.4. Stability Analysis of HKGs by Four Algorithms

#### 3.4.1. Expression Stability of the HKGs by Delta Ct Analysis

Using the Delta Ct method, gene stability was evaluated by the std-value, shown in Table 2. At stage 1, *PPIB*, a HKG we selected from RNA-seq data, demonstrated the best stability. However, it failed to be the most stable gene in other stages of all uterus caruncle samples, which *EIF3K* proved to be. Additionally, the other three HKGs selected from RNA-seq data (ie, *MORF4L1*, *HNRNPL*, and *ARF1*) did not demonstrate good stability.

#### 3.4.2. Expression Stability of the HKGs by BestKeeper Analysis

The BestKeeper algorithm determines the best HKGs based on the principle that the expression of good HKGs are highly correlated and constructs a correlative index from the results, which is then compared with the target genes to reach a conclusion whether or not the reference genes are regulated. *EIF3K* was predicted to be the most stable among stage 2, stage 3, and overall samples by the BestKeeper algorithm, with the stability value (standard deviation) of 0.46, 0.36, and 0.46, respectively (Table 3). The cut-off value for this stability measure is 1, below which all the genes are deemed to be stably expressed [9]; this was the case for all the genes investigated in stage 1 of our study. However, four and three out of the total of 17 HKGs displayed instability in stage 2 and stage 3, respectively, with stability values greater than 1. Furthermore, when every phase was considered, six HKGs were unstable.

#### 3.4.3. Expression Stability of the HKGs by NormFinder Analysis

Expression stability values determined by the NormFinder algorithm are shown in Table 4. In stage 1, *PPIB*, selected from RNA-seq data, once again showed the best stability; this is the same as the result presented in the Delta Ct method. In stage 2 and stage 3, *EIF3K* was the most stable, while *ARF1* and *RPLPO* were the least stable genes, respectively. In all samples, *EIF3K*, *POP4*, *SDHA*, and *PPIA* were the most stable, while *ARF1*, *ACTB*, *B2M*, and *RPLPO* were the least.

#### 3.4.4. Expression Stability of the HKGs by geNorm Analysis

The geNorm algorithm calculated the M Value to determine the gene stability. The genes with lowest M value have the most stable expression and stepwise exclusion of genes with highest M values will result in a combination of two genes with the highest stability [7]. The M value of each gene in every stage and all samples are shown in Table 5. In all samples, *EIF3K* and *SDHA*, with the M value of 0.940, were predicted to be the most stable genes, while *B2M* and *RPLPO* were found to be least stable, with M values of 1.758 and 1.893, respectively.

### 3.5. A Comprehensive Ranking of Four Methods

ReFinder was used to obtain an overall ranking of the selected 17 HKGs among the four methods. The result of the ReFinder algorithm are shown in Table 6. *EIF3K*, *SDHA*, *POP4*, and *PPIA* were the four most stable genes, while *ARF1*, *MORF4L1*, *HPRT1*, and *YWHAZ* were the four least stable genes across all samples.

### 3.6. Validation the Stability of HKGs with Target Genes

To validate the HKGs stability results, the most four stable genes (*EIF3K*, *POP4*, *PPIA*, and *SDHA*) and the least four stable genes (*ACTB*, *B2M*, *RPLPO*, and *ARF1*) were used to normalize the same target gene. When using the four most stable genes for normalization, the results showed that *PAG8* and *VEGFA* were expressed in uterus caruncle tissue with the lowest level at stage 1, and that they increased in stage 2, and increased significantly more in stage 3 (*p* < 0.01), where they reached the highest level (Figure 4a,c). *SPP1*, whose expression reached the highest level in stage 2, expressed the least in stage 1 and the middle in stage 3 (Figure 4e). When it comes to the four least stable genes, there were hardly any significant differences between the stages (Figure 4b,d,f). In addition, different expression trends appeared among *VEGFA* and *SPP1* when normalized to *ACTB* and *B2M*, respectively (Figure 4d,f).

A correlation analysis on the relative expression data of the four most and least stable candidate reference genes, normalized to three target genes, were performed. As shown in Figure 5, *POP4* and *PPIA* had a high correlation coefficient with an R value of 0.97. It suggested that they have extremely similar normalization capabilities. However, the R value of *ACTB* and *RPLPO* is low i.e., 0.11; this demonstrates that they have hardly normalization capabilities.

## 4. Discussion

The uterus is an important organ in the reproductive system of most female mammals; it accommodates the embryonic and fetal development of one or more embryos until birth. During mammalian pregnancy, the maternal endometrium maintains a decidua reaction, including endometrial thickening, increase of blood vessels, hypersecretion of uterine glands, and stromal cell hypertrophy. These changes in morphology and composition in the maternal uterus suggest that a genetic regulation arises from transcriptional expression. However, the HKGs that normalize the expression of target genes have not been identified during pregnancy, especially in domestic animals. Thus, it is essential to explore suitable housekeeping genes. In this study, we refer to RNA-seq data to select novel candidate HKGs of goat uterus caruncle tissue, and qRT-PCR data were used for software analysis to obtain the most suitable HKGs. Our results showed that *EIF3K*, *POP4*, *PPIA*, and *SDHA* were the most stable HKGs in the uterus regardless of pregnant or nonpregnant state.

In order to obtain the stable HKGs from RNA-seq and qRT-PCR data, we set three parameters and used four methods to rigorously screen HKGs in uterine tissues. In reference to previous studies [12,18,19], the FPKM value, the coefficient of variation, and the maximum fold change were considered in order to measure the HKGs in the RNA-seq data. Meanwhile, geNorm, NormFinder, Bestkeeper, and Delta Ct methods were used to evaluate the stability of HKGs in qRT-PCR data; specifically, a total of 22 HKGs for qRT-PCR analysis. However, only 17 HKGs were used for further study. The reason we eliminated the five HKGs is based on the notion that a Ct of less than 29 is indicative for abundant gene expression [23], which is desirable for qRT-PCR analysis. Considering the accuracy and validity of the experiment, we set up a stricter Ct value line, viz. 28.

In our results, *PPIB*, a HKG we selected from RNA-seq data, showed good stability in stage 1 referring to nonpregnant period. *PPIB* is a kind of protein coding gene associated with secretory pathway. The reason why *PPIB* demonstrated better stability than the other 16 HKGs in uterus caruncles may relate to the tissue structure of the endometrium. The uterus caruncle is the pitting structure of endometrium, which contains an abundance of epithelial cells. We predict that epithelial cells probably contribute to the secretory process. A previous study [24] has also shown that *PPIB* demonstrated the best stability in LPS-stimulated and non-stimulated *THP-1* and *K562*, which partly verified the foregoing prediction. Among the studies of HKGs performed on goats, we found a study [17] on Jianyang big-eared goats that showed that *PPIB* was the most stable gene. However, *PPIB* failed to show the best stability in the following stages. In stage 2 and stage 3, the fertilized ovum lodges in the uterus, establishing and maintaining pregnancy. A series of the aforementioned events might be more connected to genes that pertain to angiogenesis.

When it comes to samples in total stages, including stage 1, stage 2, and stage 3, *EIF3K* demonstrated the best stability. *EIF3K* (Eukaryotic translation initiation factor 3) has been suggested as one of the most stably expressed HKGs in bovine endometrium [25]. The uterus caruncle is the pitting structure of endometrium, so our results may partly validate previous research. In addition, it has been suggested that *EIF3K* is one of the most stably expressed HKGs in bovine endometrium, and is also stably expressed in bovine adipose tissue, mammary gland, muscle, liver, and goat adipose tissue [20,26]. To date, no publications were found on *EIF3K* as an unstable HKG. This may indicate the potential of *EIF3K* to serve as a stable HKG in goat uterus tissue.

It is interesting that traditional HKGs such as *ACTB* and *GAPDH* did not perform well in our research. *ACTB* was one of the four least stable genes, while *GAPDH* showed the medium stability overall. It seems to be a common conclusion that classical reference genes did not have good stability in the studies of HKGs [24,27,28,29,30]. This finding reminds us of the need to accurately validate candidate internal control genes in the study before use in gene expression studies using qRT-PCR.

When it comes to methodology in this study, we combine RNA-seq data with references to select HKGs following the use of qRT-PCR experiments to validate the selected HKGs; subsequently, the target genes of uterus caruncle tissue were used for further validation. We used four algorithms (geNorm, Delta Ct, NormFinder, and BestKeeper) to calculate the stability of HKGs. Then, ReFinder was applied to evaluate the comprehensive stability of HKGs. In general, we used all the latest methods for screening HKGs, which leads us to a relatively reliable result.

For the number of HKGs for normalization, we only chose single genes instead of gene combinations. This was determined by geNorm based on the average pairwise variation (V_n_/V_n+1_) value. As a routine, 0.15 was proposed as the cut-off value for the V value [7], below which the inclusion of an additional control gene was not required. However, all the V_n_/_n+1_ values in our study were less than 0.15 (Appendix A), indicating that increasing the number of genes did not significantly change the value of the normalization analyses. Therefore, a single top stable HKG might be a good choice for qRT-PCR assay in our study.

Previous studies have shown that *SPP1*, *VEGFA*, and *PAG8* [31,32,33] genes were the marker genes for uterus and placenta tissues. In addition, they were differentially expressed in our RNA-seq data (data was not shown) in the uterus caruncle tissues. Therefore, *SPP1*, *VEGFA*, and *PAG8* genes were selected as target genes for normalization. *SPP1* plays an important role in the regulation of maternal-fetus angiogenesis and calcium balance after implantation [34,35]. *VEGFA* (vascular endothelial growth factor A) has strong pro-angiogenic effect on embryonic vasculogenesis [36]. Early expression of *PAG* genes, including *PAG8*, in the conceptus may contribute to successful development of early pregnancy and possibly alter the mechanisms related to embryo survival, such as prostaglandin synthesis [33]. In our study, the extremely upregulated *PAG8* and *VEGFA* genes during pregnancy were shown to be significantly normalized to the most stable HKGs (*EIF3K*, *POP4*, *PPIA*, and *SDHA*). However, hardly any significance, and even severe variance, appeared when they were normalized to the four least stable HKGs (*ACTB*, *B2M*, *RPLPO*, and *ARF1*). In addition, the expression of *SPP1* demonstrated some variation between the most stable and unstable HKGs. The suitable HKGs should be carefully chosen considering the precision of qRT-PCR, as unstable reference genes will result in incorrect interpretation of expression data.

## 5. Conclusions

In our study, four different algorithms and one comprehensive index were used to analyze a panel of HKGs. All four algorithms gave consistent results. *PPIB* and *EIF3K, POP4, PPIA,* and *SDHA* proved to be the most suitable HKGs in non-pregnant stage and both non-pregnant and pregnancy stages, respectively. *ACTB*, *B2M*, *RPLPO*, and *ARF1* were the four least stable HKGs, suggesting they are not suitable for expression data normalization in studies of goat uterus caruncle tissue.

## Figures and Tables

**Figure 1 animals-13-01897-f001:**
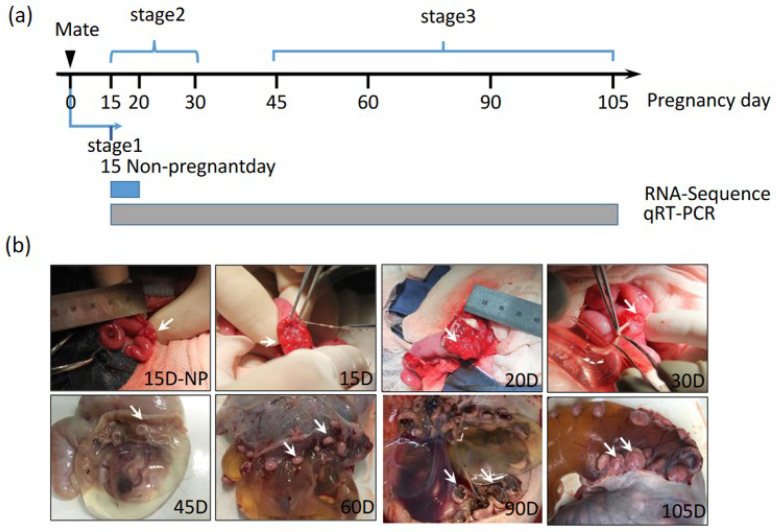
(**a**) Sampling time points (15 Non-pregnant day, 15th, 20th, 30th, 45th, 60th, 90th, and 105th of pregnancy) and stage division (stage1, stage2, and stage3) of this experiment. Blue rectangle refers to RNA-Seq performed at this stage, grey rectangle presents qRT-PCR performed at these stages. (**b**) Representative images of hircine caruncle tissues when sampling. White arrows represent positions of sample collection.

**Figure 2 animals-13-01897-f002:**
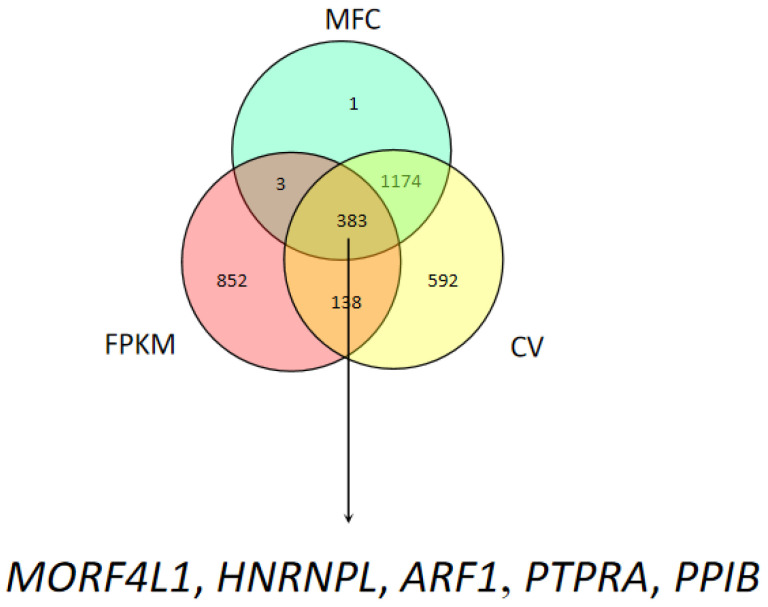
The Venn diagram analysis of overlap genes in FPKM, MFC, and CV. The arrow indicated the candidate reference genes screened by the transcriptome data.

**Figure 3 animals-13-01897-f003:**
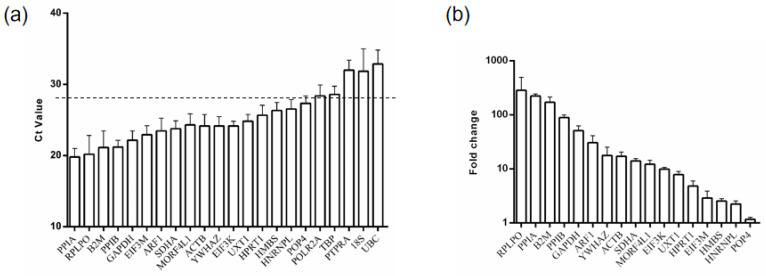
qRT-PCR of Dazu Black goat caruncle samples to analyze 22 candidate genes. (**a**) Ct values are displayed as the mean of three repeated analyses of control samples. The black line at Ct value = 28 represents the threshold set to include only candidates exhibiting moderate abundance for further analysis. (**b**) Bar chart of relative gene expression of control genes in the caruncle. Means of three control samples were normalized to the geometric mean of all gene candidates and the ΔΔCt method was used to calculate relative expression (fold change).

**Figure 4 animals-13-01897-f004:**
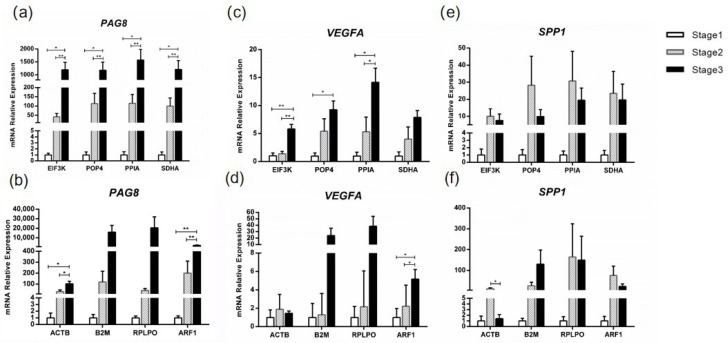
Relative expression of target genes (*PAG8*, *VEGFA*, and *SPP1*) normalized to the different reference genes across three stages. (**a**) Relative expression of *PAG8* by the most stable genes (*EIF3K*, *POP4*, *PPIA*, and *SDHA*). (**b**) Relative expression of *PAG8* by the least stable genes (*ACTB*, *B2M*, *RPLPO*, and *ARF1*). (**c**) Relative expression of *VEGFA* by the most stable genes (*EIF3K*, *POP4*, *PPIA*, and *SDHA*). (**d**) Relative expression of *VEGFA* by the least stable genes (*ACTB*, *B2M*, *RPLPO*, and *ARF1*). (**e**) Relative expression of *SPP1* by the most stable genes (*EIF3K*, *POP4*, *PPIA*, and *SDHA*). (**f**) Relative expression of *SPP1* by the least stable genes (*ACTB*, *B2M*, *RPLPO*, and *ARF1*). * *p* < 0.05, ** *p* < 0.01.

**Figure 5 animals-13-01897-f005:**
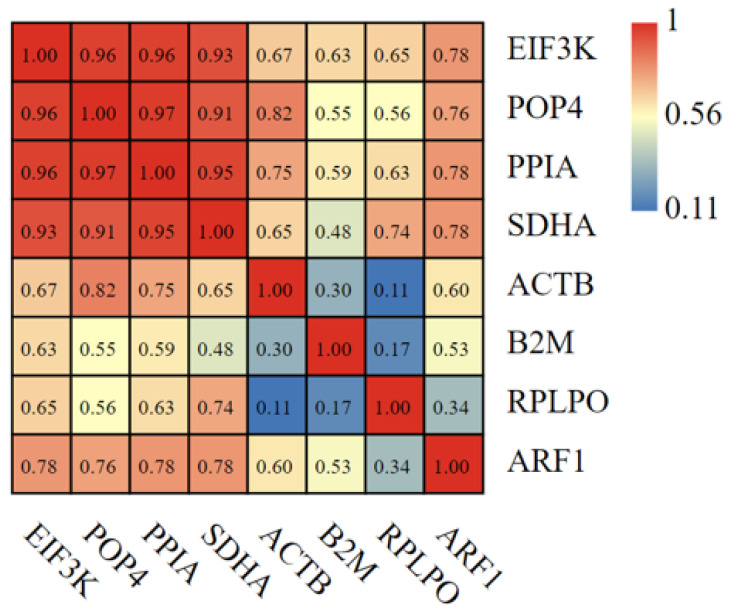
Heat map of correlation coefficients of relative gene expression levels based on different normalized HKGs. Three target genes were detected and normalized by different HKGs. The number in each color block is the correlation coefficient (R-value).

**Table 1 animals-13-01897-t001:** The information of selected candidate housekeeping genes in RNA-seq data of caruncle at non-pregnant and early pregnant days.

Gene Sources	Rank	Mean FPKM	CV	MFC	Symbol	Description
Transcriptome Data	1	378.38	0.03	1.11	*MORF4L1*	mortality factor 4 like 1 [Source:NCBI gene;Acc:102179420]
	2	190.26	0.04	1.15	*ARF1*	ADP ribosylation factor 1 [Source:NCBI gene;Acc:102185451]
	4	171.61	0.05	1.17	*HNRNPL*	heterogeneous nuclear ribonucleoprotein L [Source:NCBI gene;Acc:102180618]
	9	50.26	0.05	1.22	*PTPRA*	protein tyrosine phosphatase receptor type A [Source:NCBI gene;Acc:102184258]
	20	89.19	0.06	1.22	*PPIB*	peptidylprolyl isomerase B [Source:NCBI gene;Acc:102169391]

**Table 2 animals-13-01897-t002:** Expression stability of HKGs by Delta CT analysis.

	Stage1		Stage2		Stage3		Total	
Rank	Gene	STDEV	Gene	STDEV	Gene	STDEV	Gene	STDEV
1	*PPIB*	0.57	*EIF3K*	1.67	*EIF3K*	1.14	*EIF3K*	1.47
2	*POP4*	0.59	*UXT1*	1.73	*PPIA*	1.17	*POP4*	1.61
3	*YWHAZ*	0.61	*EIF3M*	1.85	*POP4*	1.21	*SDHA*	1.62
4	*HMBS*	0.61	*MORF4L1*	1.88	*GAPDH*	1.21	*PPIA*	1.67
5	*MORF4L1*	0.62	*HPRT1*	1.94	*SDHA*	1.23	*EIF3M*	1.70
6	*HPRT1*	0.63	*HMBS*	1.94	*HMBS*	1.25	*PPIB*	1.71
7	*EIF3K*	0.67	*PPIB*	1.97	*EIF3M*	1.25	*HMBS*	1.72
8	*SDHA*	0.68	*SDHA*	1.99	*HNRNPL*	1.30	*GAPDH*	1.75
9	*EIF3M*	0.68	*POP4*	1.99	*ACTB*	1.32	*HNRNPL*	1.77
10	*RPLPO*	0.69	*PPIA*	2.10	*YWHAZ*	1.35	*UXT1*	1.79
11	*GAPDH*	0.74	*B2M*	2.14	*UXT1*	1.37	*YWHAZ*	1.84
12	*HNRNPL*	0.76	*GAPDH*	2.22	*PPIB*	1.40	*MORF4L1*	1.87
13	*UXT1*	0.86	*HNRNPL*	2.23	*ARF1*	1.46	*HPRT1*	1.88
14	*PPIA*	0.89	*YWHAZ*	2.34	*HPRT1*	1.55	*ARF1*	2.17
15	*ARF1*	1.01	*ACTB*	2.36	*B2M*	1.81	*ACTB*	2.23
16	*B2M*	1.19	*RPLPO*	2.39	*MORF4L1*	1.82	*B2M*	2.48
17	*ACTB*	1.23	*ARF1*	2.94	*RPLPO*	3.35	*RPLPO*	2.91

**Table 3 animals-13-01897-t003:** Expression stability of HKGs by BestKeeper analysis.

	Stage1			Stage2			Stage3			Total		
Rank	Gene	Std Dev	Coeff. of Corr. [r]	Gene	Std Dev	Coeff. of Corr. [r]	Gene	Std Dev	Coeff. of Corr. [r]	Gene	Std Dev	Coeff. of Corr. [r]
1	*POP4*	0.12	0.252	*EIF3K*	0.46	0.001	*EIF3K*	0.36	0.402	*EIF3K*	0.46	0.377
2	*MORF4L1*	0.15	0.001	*RPLPO*	0.71	0.001	*PPIA*	0.42	0.601	*SDHA*	0.72	0.333
3	*HMBS*	0.18	0.198	*UXT1*	0.72	0.361	*POP4*	0.51	0.530	*PPIA*	0.76	0.427
4	*PPIB*	0.20	0.626	*EIF3M*	0.73	0.251	*SDHA*	0.57	0.539	*POP4*	0.76	0.554
5	*YWHAZ*	0.22	0.221	*SDHA*	0.78	0.035	*EIF3M*	0.58	0.396	*HMBS*	0.81	0.173
6	*EIF3M*	0.32	0.435	*HPRT1*	0.81	0.321	*GAPDH*	0.61	0.397	*PPIB*	0.82	0.093
7	*HNRNPL*	0.34	0.001	*PPIB*	0.85	0.029	*ACTB*	0.61	0.125	*YWHAZ*	0.83	0.214
8	*EIF3K*	0.36	0.506	*PPIA*	0.86	0.138	*YWHAZ*	0.64	0.001	*UXT1*	0.89	0.001
9	*SDHA*	0.38	0.776	*POP4*	0.94	0.605	*UXT1*	0.65	0.095	*HPRT1*	0.91	0.384
10	*UXT1*	0.41	0.862	*HMBS*	0.95	0.363	*HNRNPL*	0.67	0.394	*GAPDH*	0.92	0.484
11	*HPRT1*	0.42	0.769	*B2M*	0.95	0.001	*HMBS*	0.70	0.720	*EIF3M*	0.94	0.579
12	*RPLPO*	0.43	0.763	*MORF4L1*	0.96	0.708	*HPRT1*	0.72	0.001	*HNRNPL*	1.06	0.609
13	*GAPDH*	0.44	0.440	*YWHAZ*	0.99	0.483	*PPIB*	0.79	0.288	*MORF4L1*	1.26	0.512
14	*PPIA*	0.53	0.681	*GAPDH*	1.26	0.722	*ARF1*	0.88	0.547	*ARF1*	1.31	0.434
15	*ACTB*	0.73	0.001	*ACTB*	1.34	0.444	*B2M*	1.16	0.202	*ACTB*	1.44	0.001
16	*ARF1*	0.82	0.506	*HNRNPL*	1.37	0.672	*MORF4L1*	1.19	0.124	*B2M*	1.78	0.485
17	*B2M*	0.85	0.689	*ARF1*	1.72	0.438	*RPLPO*	2.83	0.455	*RPLPO*	1.974	0.335

**Table 4 animals-13-01897-t004:** Expression stability of HKGs by NormFinder analysis.

	Stage1		Stage2		Stage3		Total	
Rank	Gene	Stability Value	Gene	Stability Value	Gene	Stability Value	Gene	Stability Value
1	*PPIB*	0.181	*EIF3K*	0.726	*EIF3K*	0.408	*EIF3K*	0.527
2	*POP4*	0.227	*UXT1*	0.817	*PPIA*	0.449	*POP4*	0.845
3	*YWHAZ*	0.266	*EIF3M*	1.077	*POP4*	0.523	*SDHA*	0.893
4	*HMBS*	0.292	*MORF4L1*	1.106	*SDHA*	0.591	*PPIA*	0.989
5	*HPRT1*	0.306	*HPRT1*	1.200	*HMBS*	0.623	*EIF3M*	1.017
6	*MORF4L1*	0.323	*HMBS*	1.322	*GAPDH*	0.643	*PPIB*	1.075
7	*SDHA*	0.372	*POP4*	1.327	*EIF3M*	0.658	*HMBS*	1.094
8	*EIF3M*	0.428	*PPIB*	1.328	*HNRNPL*	0.722	*HNRNPL*	1.122
9	*EIF3K*	0.429	*SDHA*	1.382	*ACTB*	0.823	*GAPDH*	1.145
10	*RPLPO*	0.454	*PPIA*	1.577	*UXT1*	0.871	*UXT1*	1.208
11	*GAPDH*	0.473	*B2M*	1.585	*YWHAZ*	0.924	*HPRT1*	1.296
12	*HNRNPL*	0.557	*HNRNPL*	1.678	*PPIB*	0.947	*YWHAZ*	1.306
13	*UXT1*	0.707	*GAPDH*	1.694	*ARF1*	1.012	*MORF4L1*	1.326
14	*PPIA*	0.745	*YWHAZ*	1.882	*HPRT1*	1.139	*ARF1*	1.713
15	*ARF1*	0.909	*RPLPO*	1.895	*B2M*	1.512	*ACTB*	1.869
16	*B2M*	1.11	*ACTB*	1.914	*MORF4L1*	1.573	*B2M*	2.139
17	*ACTB*	1.125	*ARF1*	2.587	*RPLPO*	3.255	*RPLPO*	2.635

**Table 5 animals-13-01897-t005:** Expression stability of HKGs by geNorm analysis.

	Stage1		Stage2		Stage3		Total	
Rank	Gene	Stability Value	Gene	Stability Value	Gene	Stability Value	Gene	Stability Value
1	*HMBS*|*POP4*	0.070	*EIF3K*|*UXT1*	0.638	*HMBS*|*SDHA*	0.648	*EIF3K*|*SDHA*	0.940
2	*PPIB*	0.236	*PPIB*	0.942	*PPIA*	0.687	*PPIA*	1.006
3	*MORF4L1*	0.304	*SDHA*	1.245	*EIF3K*	0.711	*HMBS*	1.067
4	*YWHAZ*	0.327	*HMBS*	1.347	*HNRNPL*	0.779	*POP4*	1.126
5	*HNRNPL*	0.361	*PPIA*	1.393	*POP4*	0.824	*PPIB*	1.178
6	*EIF3K*	0.395	*EIF3M*	1.440	*EIF3M*	0.855	*EIF3M*	1.237
7	*HPRT1*	0.423	*POP4*	1.500	*GAPDH*	0.889	*UXT1*	1.306
8	*RPLPO*	0.445	*MORF4L1*	1.537	*ACTB*	0.936	*HNRNPL*	1.368
9	*EIF3M*	0.468	*B2M*	1.590	*YWHAZ*	0.967	*GAPDH*	1.416
10	*SDHA*	0.496	*HPRT1*	1.649	*UXT1*	0.992	*MORF4L1*	1.459
11	*GAPDH*	0.522	*HNRNPL*	1.737	*PPIB*	1.024	*YWHAZ*	1.495
12	*UXT1*	0.561	*GAPDH*	1.809	*ARF1*	1.057	*HPRT1*	1.533
13	*PPIA*	0.598	*YWHAZ*	1.886	*HPRT1*	1.099	*ARF1*	1.601
14	*ARF1*	0.640	*ACTB*	1.937	*MORF4L1*	1.169	*ACTB*	1.661
15	*B2M*	0.705	*RPLPO*	1.988	*B2M*	1.232	*B2M*	1.758
16	*ACTB*	0.766	*ARF1*	2.100	*RPLPO*	1.482	*RPLPO*	1.893

**Table 6 animals-13-01897-t006:** Comprehensive rankings calculated using the the ReFinder method.

	Stage1		Stage2		Stage3		Total	
Rank	Gene	Score	Gene	Score	Gene	STDEV	Gene	STDEV
1	*POP4*	1.41	*EIF3K*	1.00	*EIF3K*	1.41	*EIF3K*	1.00
2	*PPIB*	1.86	*UXT1*	1.68	*PPIA*	2.21	*SDHA*	2.45
3	*HMBS*	2.63	*EIF3M*	3.98	*SDHA*	2.99	*POP4*	3.16
4	*YWHAZ*	3.87	*PPIB*	4.74	*POP4*	3.57	*PPIA*	4.43
5	*MORF4L1*	3.94	*MORF4L1*	5.83	*HMBS*	4.26	*PPIB*	4.56
6	*HPRT1*	7.00	*HPRT1*	6.37	*GAPDH*	5.83	*HMBS*	6.09
7	*EIF3K*	7.71	*HMBS*	6.82	*EIF3M*	6.44	*EIF3M*	6.62
8	*EIF3M*	8.11	*SDHA*	7.14	*HNRNPL*	7.52	*UXT1*	7.00
9	*SDHA*	8.63	*POP4*	8.63	*ACTB*	8.45	*GAPDH*	8.97
10	*HNRNPL*	8.82	*PPIA*	8.80	*YWHAZ*	9.69	*HNRNPL*	9.39
11	*RPLPO*	9.97	*B2M*	9.59	*UXT1*	10.22	*YWHAZ*	9.87
12	*GAPDH*	11.49	*RPLPO*	11.77	*PPIB*	12.24	*HPRT1*	11.68
13	*UXT1*	13.24	*HNRNPL*	13.16	*ARF1*	13.24	*MORF4L1*	12.45
14	*PPIA*	13.74	*GAPDH*	13.21	*HPRT1*	13.47	*ARF1*	14.24
15	*ARF1*	15.24	*YWHAZ*	13.74	*B2M*	15.24	*ACTB*	14.47
16	*B2M*	16.24	*ACTB*	14.98	*MORF4L1*	15.74	*B2M*	16.00
17	*ACTB*	16.48	*ARF1*	17.00	*RPLPO*	17.00	*RPLPO*	17.00

## Data Availability

The datasets used and/or analyzed during the current study are available from the corresponding author on reasonable request.

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
