# Peer review of "Screening of Candidate Housekeeping Genes in Uterus Caruncle by RNA-Sequence and qPCR Analyses in Different Stages of Goat (Capra hircus)"

_animals, 2023, doi:10.3390/ani13121897_

Round 1

Reviewer 1 Report

Dear Author

the article may be corrected as suggested

Author Response

Dear reviewer,

Thank you for all your revisions and suggestions.  We have revised them all and expanded the content in the discussion. Please see the revised manuscript for details. 

Reviewer 2 Report

Abstract (line 11) and introduction (lines 34-35): critical or necessary? Please compare the two words.

Please add a space before the round brackets, where it is lacking.

Please check the spaces between words on the figure legend. The correct version is present in the “original images” file.

Lines 129-130: please explain why “a candidate HKG should have a relatively high FPKM value,”.

Supplementary materials are not available.

Line 144: annealing or annealing and extension?

Table 1: gene symbols must be written in italic.

Only Table 1 is cited in the manuscript.

Figure 6 is not cited in the text. Moreover, why you reported here 4 genes (EIF3K, POP4, PPIA, SDHA) as the most stable genes and don’t include PPIB, one of the selected HKGs as indicated in the conclusions and in other parts of the manuscript?

Author Response

Dear reviewer,

Thank you for all your suggestions, and we have revised and replied one by one. Please refer to the attachment for details

Round 2

Reviewer 2 Report

Simple summary:

1. caruncle > caruncles

2. goat > goats

3. house keeping genes(HKGs) > housekeeping genes (HKGs)

4. Carpra Hircus> Capra hircus

5. the suitable > suitable

Line 141: quantity > quantitative OR USE "qRT-PCR"

Table 1: Non-pregnant > non-pregnant

line 395: EIF3K or EIF3K, POP4, PPIA, SDHA?

Figure S2: RT-qPCR > qRT-PCR

Author Response

Dear reviewer, 

Thanks for your prompt reply and meticulous review comments. We have carefully checked all the mistakes and corrected them according to your suggestion. Please refer to the attachment for details of modification. Just one  modification in supplementary figures may not be displayed in the attachment, because only one word file can be uploaded. 

Bests, 

Nanjian Luo
